# Research on the Wood Density Measurement in Standing Trees through the Micro Drilling Resistance Method

Jianfeng Yao [1,2,3], Yabin Zhao [1], Jun Lu [4,*], Hengyuan Liu [1], Zhenyang Wu [1], Xinyu Song [2,3,5] and Zhuofan Li [2,3,6]

1    College of Computer and Information Technology, Xinyang Normal University, Xinyang 464000, China; yaojf@xynu.edu.cn (J.Y.); zhaoyb@xynu.edu.cn (Y.Z.); liuhy@xynu.edu.cn (H.L.); wuzy@xynu.edu.cn (Z.W.)
2    Henan Dabieshan National Field Observation and Research Station of Forest Ecosystem, Zhengzhou 450046, China; xysong88@xynu.edu.cn (X.S.); lizhuofan@mails.ccnu.edu.cn (Z.L.)
3    Xinyang Academy of Ecological Research, Xinyang 464000, China
4    Research Institute of Forest Resource Information Techniques, Chinese Academy of Forestry, Beijing 100091, China
5    College of Mathematics and Statistics, Xinyang Normal University, Xinyang 464000, China
6    College of Tourism, Xinyang Normal University, Xinyang 464000, China
*    Correspondence: junlu@caf.ac.cn

**Abstract:** To achieve a micro-destructive and rapid measurement of the wood density of standing trees, this study investigated the possibility of the unified modeling of multiple tree species, the reliability of the micro drilling resistance method for measuring wood density, the relationship between drilling needle resistance and wood density, and whether moisture content has a significant impact on the model. First, 231 tree cores and drill resistance data were sampled from *Pinus massoniana*, *Cunninghamia lanceolate*, and *Cryptomeria fortunei*. The basic density and moisture content of each core were measured, and the average value of each resistance data record was calculated. Second, the average drill resistance, the natural logarithm of average drill resistance, and absolute moisture content were used as independent variables, while the basic wood density was used as the dependent variable. Third, the total model of the three tree species and sub-model for each tree species were established through a stepwise regression method. Finally, the accuracy of each model was compared and analyzed with that of using the average basic density of each tree species as an estimated density. The estimated accuracy of the total model, sub model, and average wood density modeling data were 90.070%, 93.865%, and 92.195%, respectively. The results revealed that the estimation accuracy of the sub-model was 1.670 percentage points higher than that of the average wood density modeling data, while the estimation accuracy of the total model was 2.125 percentage points lower than that of the average wood density modeling data. Additionally, except for *Cryptomeria fortunei*, the natural logarithm of drill resistance significantly influenced the wood density model at a significance level of 0.05. Moreover, moisture content significantly affected the total model and sub-models of *Pinus massoniana* at a significance level of 0.05. The results indicated the feasibility of using the micro-drilling resistance method to measure the wood density of standing trees. Moreover, the relationship between wood density and drill resistance did not follow a linear pattern, and moisture content slightly influenced the drill needle resistance. Furthermore, the establishment of a mathematical model for each tree species was deemed essential. This study provides valuable guidance for measuring the wood density of standing trees through the micro-drilling resistance method.

**Keywords:** mathematical model; micro-drilling resistance method; standing trees; wood density

## 1. Introduction

Wood density is the mass of wood per unit volume at a specific moisture content. This parameter is considered the most reliable predictor of wood quality [1]. Wood density exhibits a strong correlation with other wood qualities, such as strength and stiffness,

and plays a significant role in determining wood suitability for different end uses [2]. Wood density is influenced by both genetic factors [3–5] and the growth environment of trees [6–9]. Therefore, wood density serves as a vital evaluation parameter in studies related to tree genetic breeding and forest management methods. With the development of human society, the demand for wood products has gradually increased [10–13], while the availability of wood resources has significantly decreased. Improving the efficiency of timber use emerges as an effective measure to address the imbalance between wood supply and demand. Accurate measurement of wood density forms the basis for enhancing the efficiency of timber use. Forest managers and wood processors use wood density to effectively match raw materials with final products [14]. Additionally, wood density is a vital important factor in forest carbon measurement [15–18]. Improving the accuracy of measuring wood density can contribute to enhanced precision in estimating forest carbon. Wood density exhibits significant variability among individual trees in forests [19–21]. Therefore, promoting sustainable human development requires the advancement of rapid, accurate, and non-destructive methods for assessing wood density in standing trees.

The traditional approach for measuring wood density of standing trees is the volume method [22]. This method first requires sampling wood samples from standing trees, then measuring the volume of fresh wood in the laboratory, and finally drying the samples to measure their absolute dry density. Although the volume method can accurately measure wood density, its dependence on sampling from the tested object leads to significant specimen damage. Moreover, the sampling and measurement processes are time-consuming and labor-intensive. The X-ray method indirectly measures wood density based on the intensity of X-rays absorbed by substances with different densities [23,24]. Moreover, this method accurately measures the density of wood in small areas and enables the assessment of average density in tree rings, early wood density, and late-wood density. However, X-ray instruments for measuring density are expensive, and the method also requires sampling from the tested object. Therefore, the X-ray method is costly, time-consuming, and labor-intensive. In contrast, the Pilodyn method indirectly measures wood density by inserting a fine needle with a fixed specification into the wood using a preloaded spring and then gauging the depth of the needle's penetration into the wood [25,26]. Despite the convenience of this instrument, it only measures the average density of the outer wood surface. The micro drill resistance instrument uses a motor to control the constant-speed penetration of the drill needle into wood. The drill resistance is positively correlated with wood density and the instruments can measure wood density indirectly [27–33]. The resistance drilling method has considerable advantages over other methods, including minimal tree damage, faster operation, and higher measurement sensitivity, making it a highly promising method for measuring wood density [22].

Currently, scholars have mainly used linear models to investigate the relationship between drill resistance and wood density. For example, Rinn established a linear model with a correlation of $r^2 = 0.943$ between drill resistance and wood density [27]. Isik et al. revealed strong correlations between average drilling resistance values and wood density, indicating strong genetic control at the family level. However, individual phenotypic correlations were observed to be relatively weak [34]. Downes et al. found determination coefficients of the linear models between the average drill resistance and wood density of each tree in various plots ranging from 0.662 to 0.868 [35]. Due to the significant differences in the parameters and determination coefficients among various linear models, the universality of these models was poor. Therefore, researchers needed to establish a mathematical model for every tree species, and the modeling workload was enormous. In addition, the scatter plots of drill needle resistance and wood density were relatively scattered, and some data points had a large distance from the fitting curve. Therefore, the reliability of this method needs further verification.

Owing to significant differences in the parameters and determination coefficients across various linear models, the universal applicability of linear models is limited. Presently, researchers typically need to establish specific linear models for different tree species

when using micro drill resistance instruments to measure wood density. Establishing a unified mathematical model for multiple tree species poses a challenge. Additionally, the relationship between drill resistance and wood density does not follow a linear pattern. Numerous scholars have shown that the relationship between various wood mechanical properties and wood density can be expressed by a *k*-th parabolic equation:

$$s = \alpha \rho^k \tag{1}$$

where *s* represents a wood strength value; *α* denotes the proportional constant; and *k* is the density index, shaping the relationship curve between wood strength and wood density. Certain wood strength properties exhibit exponential variations with changes in wood density. For example, the density index of flexural strength is 1.25, while the density index of transverse compressive strength and hardness is 2.25 [36]. Therefore, an exponential variation may occur between drill resistance and wood density.

Additionally, the moisture content of wood has a certain influence on its strength [37–39]. Therefore, the moisture content also affects drill resistance. Lin et al. found that drill resistance values typically decreased with decreasing moisture content, transitioning from a water-saturated condition to air-dried status for *Taiwania cryptomerioides* lumber [37]. Sharapov et al. reported that the impact of moisture content on drill needle resistance and drill feeding force depended on the rotational speeds and rates of the drill [38]. Ukrainetz et al. found that density prediction by drill resistance was influenced by tree moisture content [39]. Owing to significant differences in moisture content among different tree species, locations, and times, further research is needed to investigate the impact of moisture content on measuring the basic density of standing trees.

In order to further investigate the possibility of the unified modeling of multiple tree species, the reliability of the micro drilling resistance method for measuring wood density, the relationship between drilling needle resistance and wood density, and whether moisture content has a significant impact on the model, this paper further studies the micro drilling resistance measurement method for wood density. The average drill resistance, the natural logarithm of average drill resistance, and absolute moisture content were used as independent variables, while the basic wood density served as the dependent variable. Total models for multiple tree species and sub models for each tree species were established with stepwise regression. The accuracy and standard deviation of the estimated results of the total model, sub models, and estimated results were compared with the average basic density of the building data.

## 2. Materials and Methods

### 2.1. Core Sampling and Drill Resistance Measurements

In October 2023, increment cores and drill resistance data were obtained from 10 plots of *Pinus massoniana* and 10 mixed forests plots of *Cunninghamia lanceolate* and *Cryptomeria fortunei* in Jigongshan Nature Reserve (114°01′–114°06′ E, 31°46′–31°52′ N), Xinyang City, Henan Province, China. Each plot measured 20 m × 20 m and was divided into 4 small quadrants of 10 m × 10 m. In each quadrat, one dominant tree, one moderate tree, and one suppressed tree were selected as test trees. Among the 40 *Pinus massoniana* quadrants, the number of *Pinus massoniana* trees in three quadrants were below 3. Therefore, no test tree was selected in these three quadrants, and the number of test *Pinus massoniana* trees was 111. In 10 mixed forests of *Cunninghamia lanceolate* and *Cryptomeria fortunei*, 60 *Cunninghamia lanceolate* trees and 60 *Cryptomeria fortunei* trees were selected as test trees. Each test tree has a core sampled in the north–south direction using an increment corer with an inner diameter of 5.15 mm at a height of 1.3 m. Moreover, the drill needle resistance was sampled using a Resistograph 650-s (Rinntech Company, Heidelberg, Germany) in the same direction. To minimize the difference in wood density between the cores and the drilled wood and prevent the overlapping of the sampling paths of the increment corer and Resistograph 650-s, the distance between the sampling points of the two instruments was maintained at 3–5 cm. The sampling method is shown in Figure 1.

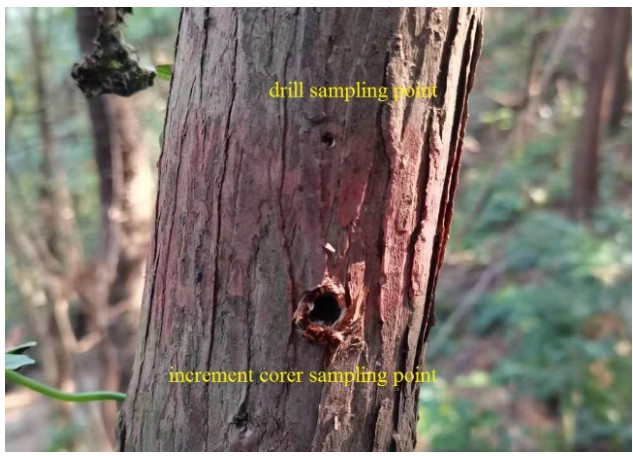

**Figure 1.** Schematic of sampling method.

*2.2. Basic Density and Moisture Content Measurements of the Cores*

After extracting the core from the increment corer, we first removed the bark at both ends of the core. Subsequently, the length ($l$) of the core was measured using a ruler, and the mass ($m_0$) of the core was calculated using an electronic balance. The diameter of the core was 5.15 mm. The volume of fresh cores was calculated using the following formula.

$$V = 3.141 \times (5.15/2)^2 l \tag{2}$$

where $V$ represents the volume of fresh cores (cm$^3$), and $l$ denotes the length of the fresh core (cm).

In the laboratory, the cores were first fixed in the wood core groove and then subjected to baking in an oven until they reached an absolutely try state. Finally, the absolute dry mass of every core ($m_1$) was measured using an electronic balance. The basic density of the core was calculated using the following formula.

$$\rho = m_1 / V \tag{3}$$

where $\rho$ is the basic density of cores (g/cm$^3$), and $m_1$ is the mass of the dry core (g). The moisture content of the cores was calculated using the following formula.

$$w = (m_0 - m_1)/m_1 \tag{4}$$

where $w$ denotes the moisture content of cores (%).

*2.3. Calculation Method for Average Drill Resistance of Each Test Tree*

The drill geometry of the Resistograph is shown in Figure 2 [29].

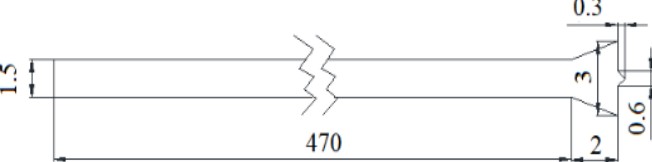

**Figure 2.** Drill geometry of Resistograph.

The width of the tip was twice the diameter of the drill shaft, and the drill resistance was mainly concentrated on the tip. However, some friction still occurred between the needle shaft and the drilling hole. The drill needle penetrated the tree after advancing ~1 cm in the drill needle socket. During this 1 cm displacement, the drill needle remained unloaded, both the drill bit resistance and drill needle shaft friction were negligible (~0), and the primary source of drill resistance was mainly the energy consumed by the motor.

As the drill needle advanced ~2 cm out of the tree, it withdrew from the tree. During the withdrawal phase, as the drill needle exited the tree, the resistance of the drill bit was reduced to 0, and the remaining drill resistance was mainly attributed to the drill shaft friction and the energy consumed by the motor. Therefore, the difference between the average drill resistance after the drill exited the tree and the average drill resistance before the drill penetrated the tree represents the friction of the drill shaft. We assumed that the drill shaft friction was proportional to the drilling depth. Therefore, the resistance of each data point after eliminating drill shaft friction is calculated using the following formula.

$$F_1 = F_0 - (f_0 - f_i)L_0/L \tag{5}$$

where $F_1$ is the drill resistance after removing drill shaft friction, $F_0$ denotes the original drill resistance, $f_0$ represents the average drill resistance after the drill exited the tree, $f_i$ is the average drill resistance before the drill penetrated the tree, $L_0$ is the real-time drilling depth, $L$ denotes the total drilling depth. The complete drill needle resistance curve and the schematic of removing the drill shaft friction are shown in Figure 3.

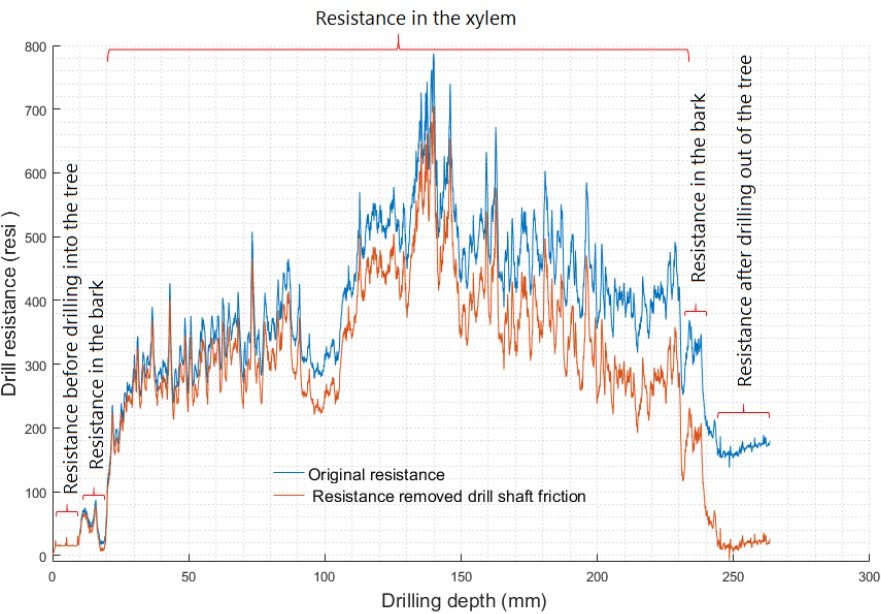

**Figure 3.** Drill resistance curve.

### 2.4. Statistical Analysis, Modeling, and Testing

The average drill resistance, the basic wood density, and the absolute moisture content of each tree were considered as one data record, resulting in 231 data records. First, the box plot method was used to eliminate abnormal data for each parameter of each tree species with R language. Subsequently, 2/3 of the data for each tree species were randomly selected as the modeling dataset. The average drill resistance, the natural logarithm of average drill resistance, and absolute moisture content were used as independent variables, while the basic wood density served as the dependent variable. The basic density regression models for the three tree species and each tree species were established through the stepwise linear regression method. Finally, the remaining 1/3 of the data was used as the test dataset. The estimated accuracy and standard deviation of each model were calculated using Formulas (6) and (7):

$$\xi = \frac{\sum_{i=1}^{n}\left(1 - \frac{|\hat{y}_i - y_i|}{y_i}\right)}{n} \tag{6}$$

$$\sigma = \sqrt{\frac{\sum_{i=1}^{n}(\hat{y}_i - y_i)^2}{n-2}} \tag{7}$$

where $\xi$ is the estimated accuracy, $n$ is the number of the total test data, $\hat{y}_i$ is the estimated value of the $i$th data, $y_i$ is the true value of the $i$th data, $\sigma$ is the standard deviation.

The measurement accuracy and estimated standard deviation of each model were compared. To verify the practical value of micro drill resistance method for measuring basic wood density, the estimated results of micro drill resistance method were compared with the average wood density of the modeling dataset for each tree species.

## 3. Results

### 3.1. Results of Removing Outliers

The confidence level of the box plot was set at 95%. Figure 4 shows the box plots of the average drill resistance, the basic wood density, and the absolute moisture content of each tree.

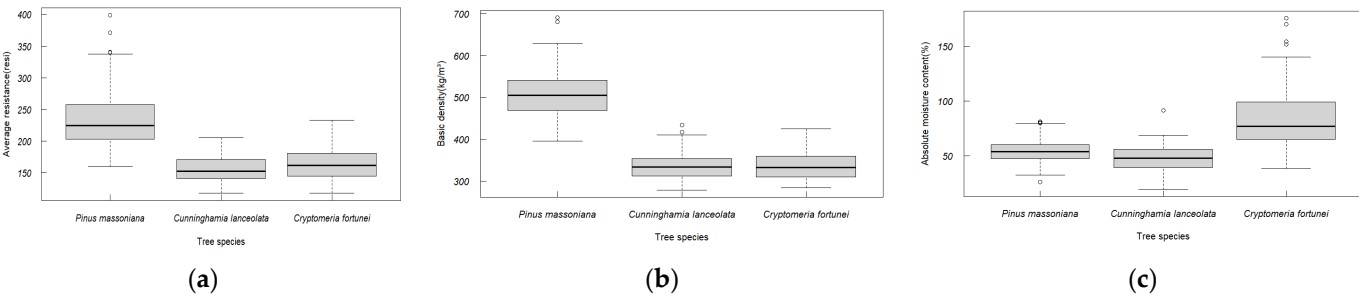

**Figure 4.** Box plots. (**a**) Average drill resistance. (**b**) Basic wood density. (**c**) Moisture content.

The data with a confidence level above 95% were considered abnormal. Figure 4 reveals 18 outliers. Owing to the presence of two outliers in the same record data, 17 data records were deleted. The overview of the data is shown in Table 1.

**Table 1.** Overview of the data.

| Species | Number of Original Data Records | Number of Data Records after Removing Outliers |
|---|---|---|
| *Pinus massoniana* | 111 | 101 |
| *Cunninghamia lanceolate* | 60 | 57 |
| *Cryptomeria fortunei* | 60 | 56 |
| Total | 231 | 214 |

### 3.2. Modeling Results

3.2.1. Mathematical Model for the Three Tree Species

The complete set of modeling data was used to establish a linear model relating basic wood density, the average drill resistance, the natural logarithm of the average drill resistance, and the absolute moisture content.

Owing to the highest $p$-value of the average drill resistance and its significant impact on the model at a significance level of 0.05, the average drill resistance can be removed from the independent variables. After excluding the average resistance of the drill needle, the parameters of the linear model for the basic density of the wood are shown in Table 2.

The $p$-values of each parameter were all below 0.05, indicating that each parameter significantly influenced the basic wood density at a significance level of 0.05 (Table 2). Therefore, the linear model between basic wood density, the natural logarithm of the average drill resistance, and the absolute moisture content was used as the mathematical model for measuring the basic wood density of these three tree species. The adjusted coefficient of determination for this model was 0.779.

**Table 2.** *p*-values of each parameter in the linear model for basic wood density, the natural logarithm of the average drill resistance, and absolute moisture content.

| Parameter | Parameter Value | *p*-Value |
|---|---|---|
| Intercept | −1279.480 | <0.001 |
| The natural logarithm of average drill resistance | 335.320 | <0.001 |
| Absolute moisture content | −97.400 | <0.001 |

### 3.2.2. Mathematical Model for Each Tree Species

Mathematical Model for *Pinus massoniana*

The modeling data of *Pinus massoniana* were used to develop a linear model correlating basic wood density with the average drill resistance, the natural logarithm of the average drill resistance, and the absolute moisture content.

Owing to the highest *p*-value of the average drill resistance and its negligible influence on the model at a 0.005 significance level, the average drill resistance can be excluded from the independent variables. After removing the average resistance of the drill needle, the parameters of the linear model for basic wood density are shown in Table 3.

**Table 3.** *p*-values of each parameter in the linear model for *Pinus massoniana*, involving basic wood density, the natural logarithm of the average drill resistance, and absolute moisture content.

| Parameter | Parameter Value | *p*-Value |
|---|---|---|
| Intercept | −656.000 | <0.001 |
| The natural logarithm of average drill resistance | 229.190 | <0.001 |
| Absolute moisture content | −147.320 | 0.001 |

The *p*-values of each parameter were all below 0.05, indicating that each parameter considerably influenced the basic wood density at a significance level of 0.05 (Table 3). Therefore, the mathematical model used to calculate the basic wood density of *Pinus massoniana* incorporated a linear relationship among the basic wood density, the natural logarithm of the average drill resistance, and the absolute moisture content. The adjusted coefficient of determination for this model was 0.588.

Mathematical Model for *Cunninghamia lanceolate*

The modeling data of *Cunninghamia lanceolate* were used to construct a linear model correlating basic wood density with the average drill resistance, the natural logarithm of the average drill resistance, and the absolute moisture content.

Using stepwise regression to sequentially remove variables with the least significant impact on the model, the parameters of the linear model for basic wood density are shown in Table 4.

**Table 4.** *p*-values of each parameter in the linear model for *Cunninghamia lanceolate*, involving basic wood density and the natural logarithm of the average drill resistance without intercept.

| Parameter | Parameter Value | *p*-Value |
|---|---|---|
| The natural logarithm of average drill resistance | 66.958 | <0.001 |

The *p*-values of the natural logarithm of average drill resistance were less than 0.05, indicating that the natural logarithm of average drill resistance significantly affected the basic wood density at a significance level of 0.05 (Table 4). Therefore, the mathematical model for calculating the basic wood density of *Cunninghamia lanceolate* incorporated the linear relationship between basic wood density and the natural logarithm of the average drill resistance. The adjusted coefficient of determination of this model was 0.994.

Mathematical Model for *Cryptomeria fortunei*

The modeling data of *Cryptomeria fortunei* were used to establish a linear model correlating basic wood density with the average drill resistance, the natural logarithm of the average drill resistance, and the absolute moisture content.

After removing variables that have no significant impact on the model with stepwise regression, the parameters of the linear model for the basic wood density are shown in Table 5.

**Table 5.** *p*-values of each parameter in the linear model for *Cryptomeria fortunei*, involving basic wood density and average drill resistance.

| Parameter | Parameter Value | *p*-Value |
|---|---|---|
| Intercept | 208.746 | <0.001 |
| Average drill resistance | 0.791 | <0.001 |

The *p*-values of average drill resistance were below 0.05, indicating a significant influence of average drill resistance on basic wood density at a significance level of 0.05 (Table 5). Therefore, the mathematical model for calculating the basic wood density of *Cryptomeria fortunei* incorporated the linear relationship between basic wood density and the average drill resistance. The adjusted coefficient of determination for this model was 0.347.

*3.3. Test Results*

Using the test data set, the average accuracy, standard deviation of the total model estimate, sub-tree model estimate, and average basic density for each tree species in the modeling data were calculated and used as the estimated basic density for the respective tree species. The test results are shown in Table 6.

**Table 6.** Test results.

| Spieces | Total Model | | Sub Model | | Average Basic Density of Each Tree Species | |
|---|---|---|---|---|---|---|
| | Estimated Standard Error/(kg·m$^{-3}$) | Mean Estimated Accuracy (%) | Estimated Standard Error/(kg·m$^{-3}$) | Mean Estimated Accuracy (%) | Estimated Standard Error/(kg·m$^{-3}$) | Mean Estimated Accuracy (%) |
| *Pinus massoniana* | 58.646 | 91.401 | 47.393 | 93.248 | 47.669 | 92.639 |
| *Cunninghamia lanceolate* | 46.505 | 88.491 | 27.062 | 93.263 | 31.138 | 92.260 |
| *Cryptomeria fortunei* | 44.238 | 89.337 | 18.087 | 95.540 | 36.172 | 91.360 |
| Total | 50.599 | 90.070 | 35.639 | 93.865 | 39.776 | 92.195 |

The mathematical model for each tree species exhibited the highest estimation accuracy, while the total model for the three tree species featured the lowest estimation accuracy (Table 6).

## 4. Discussion

Higher wood density indicates stronger wood strength and an increased energy demand for drilling through the wood. Therefore, drill resistance serves as an estimate for wood density. Currently, researchers have mainly used linear models to estimate wood density based on drill resistance. Tomczak et al. measured the radial basic density of nine oak trees with an increment corer and IML (Instrumenta Mechanik Labor, Australia) power drill. The results indicated that the determination coefficient of the linear model between the average drill resistance and wood density was 0.396 [40]. Close relationships existed in the regression model between the amplitude of the drilling resistance and wood density in previous research on trees in a breeding program ($R^2 > 0.60$) [41], lumber and the linear regression model for agarwood ($R^2 = 0.25$) [42]. However, owing to the exponential

relationship between wood strength and wood density, the relationship between drill resistance and wood density does not follow a linear pattern. The results of this study revealed that except for *Cryptomeria fortunei*, the natural logarithm of drill resistance significantly influenced the wood density model. Figure 5 shows a comparison between the linear and logarithmic fitting curves of the average drill resistance and wood density on the modeling dataset. Table 7 displays the linear and logarithmic fitting equations of the average drill resistance and wood density.

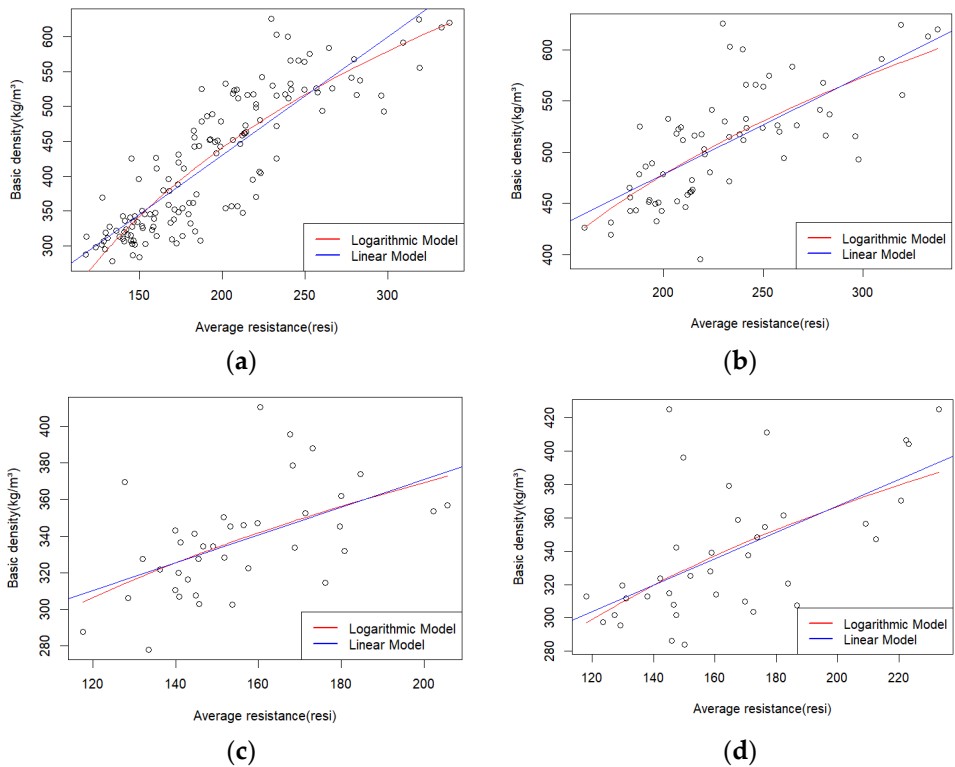

**Figure 5.** Comparison between the linear and logarithmic fitting curves of the average drill resistance and wood density. (**a**) Total modeling data. (**b**) *Pinus massoniana*. (**c**) *Cunninghamia lanceolate*. (**d**) *Cryptomeria fortunei*.

**Table 7.** Linear and logarithmic fitting equations of the average drill resistance and wood density.

| Species | Linear Model | | Logarithmic Model | |
|---|---|---|---|---|
| | Equation | Adjusted $R^2$ | Equation | Adjusted $R^2$ |
| *Pinus massoniana* | y = 285.499 + 0.965x | 0.506 | y = −766.910 + 234.970lnx | 0.521 |
| *Cunninghamia lanceolate* | y = 219.332 + 0.759x | 0.256 | y = −283.580 + 123.220lnx | 0.270 |
| *Cryptomeria fortunei* | y = 208.746 + 0.791x | 0.347 | y = −335.230 + 132.510lnx | 0.336 |
| Total | y = 91.366 + 1.692x | 0.733 | y = −1370.740 + 341.840lnx | 0.746 |

The logarithmic models exhibited higher fitting accuracy than linear models, except for *Cryptomeria fortunei* (Table 7).

Moisture content significantly influences the mechanical properties of wood, thereby affecting drill resistance. The results of this study revealed that moisture content significantly influenced the total model and sub-model of *Pinus massoniana*. Thus, measuring the moisture content of wood is challenging. Therefore, when using the micro drill resistance method for measuring wood density, most users do not measure the moisture content of wood. Therefore, to measure the wood density of standing trees, it is advisable to sample

the drill resistance in a consistent environment to minimize the influence of moisture content on drill resistance.

From a macro perspective, higher wood density corresponded to greater drill resistance. However, the mathematical model established using data from multiple tree species featured lower test accuracy, even lower than predicting the density of each tree species based on the average basic density of each tree species. This difference may be attributed to the relationship between wood strength and density, which can be influenced by fiber length, cell wall thickness, resin content, and other factors. Thus, species with similar densities can still differ from each other in wood anatomical structure and resin content. Therefore, establishing a mathematical model for each tree species is vital.

The sub-model for each tree species a exhibited higher estimation accuracy than the average value of each tree species used as the density estimation value. This confirmed the feasibility of using the micro drilling resistance method to measure the wood density of standing trees. However, the estimated accuracy of the total test data for each sub-model was only 1.670 percentage points higher than the average of each tree species used as the density estimate. This difference may be attributed to the following reasons. First, the thin and long structure of the drill needle led to a significant vibration amplitude during high-speed rotation, resulting in the formation of noise signals in the resistance measurement. Second, the operator's actions, such as breathing, trembling, and movement, can introduce vibrations in the micro drill resistance meter, thereby affecting the drill resistance measurements. Future research should focus on improving the mechanical strength of the drill needle to reduce vibrations and designing a bracket to stabilize the micro drill resistance instrument and mitigate the impact of the operator's movements on the accuracy of drill resistance measurements.

## 5. Conclusions

From the research and analyses conducted by the authors, the following conclusions are drawn:

1. The use of the micro drilling resistance method for measuring the wood density of standing trees was feasible;
2. The relationship between wood density and drill resistance did not follow a linear pattern; in some tree species, this relationship exhibited a logarithmic pattern;
3. The establishment of a mathematical model for each tree species was considered essential.

This study provides valuable guidance for measuring the wood density of standing trees using the micro-drilling resistance method. However, owing to the inconsistent effect of the natural logarithm of average drill resistance and moisture content on some sub-models, further research in these two aspects is required in the future.

**Author Contributions:** Conceptualization, J.Y. and J.L.; methodology, J.Y., Y.Z., H.L. and Z.W.; data sampling and processing, J.Y.; validation, Y.Z.; formal analysis, Z.L.; investigation, J.Y. and Z.L.; resources, J.Y. and Z.L.; data curation, Y.Z., H.L. and Z.W.; writing—original draft preparation, J.Y.; writing—review and editing, J.Y. and J.L.; visualization, J.Y. and Z.L.; supervision, X.S. and J.Y.; project administration, J.Y.; funding acquisition, J.L. and J.Y. All authors have read and agreed to the published version of the manuscript.

**Funding:** This research was funded by the National Key R&D Program of China (Grand No. 2021YFD2200404) and the Natural Science Foundation of Henan Province (232300421167), the Key Scientific Research Projects of Universities in Henan Province (22A220002), the Xinyang Academy of Ecological Research Open Foundation (2023XYQN04), the Xinyang Academy of Ecological Research Open Foundation (2023XYZD02), the Natural Science Foundation of Henan Province (222300420274), the Academic Degrees & Graduate Education Reform Project of Henan Province (2021SJGLX057Y), and the Postgraduate Education Reform and Quality Improvement Project of Henan Province (YJS2023SZ23).

**Data Availability Statement:** No new data were created or analyzed in this study. Data sharing is not applicable to this article.

**Acknowledgments:** Shouzheng Tang, an academician of the Chinese Academy of Forestry, provided guidance on topic selection; Xiangdong Lei, a researcher of the Chinese Academy of Forestry, provided the Resistograph 650-SC instrument.

**Conflicts of Interest:** The authors declare no conflicts of interest.

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
