# Peer review of "Research on the Wood Density Measurement in Standing Trees through the Micro Drilling Resistance Method"

_forests, doi:10.3390/f15010175_

Round 1
Reviewer 1 Report
Comments and Suggestions for Authors
Line 64: Please explain the "volume method". Do you mean destructively measuring the weight and volume of an increment core or other small piece of wood? Perhaps using a different descriptor may help if that is what you mean.
Line 78- you may wish to point out that the resistance drill method is mainly used for detecting defects in a standing tree or structure but can also be used to estimate density as you go on to describe.
Line 144 should be "quadrants" not quadrats.
Line 152- give manufacturer and city location for Resistograph 650-s
Line 162-163 this sentence does not make sense.
Section 3.2. The language is highly repetitive here. This section can be greatly shortened with just the results of the model fits after you walk through it for the three species set.
Line 372 & Conclusion #3, this doesn't make sense. How can you control the moisture content of a standing tree?
Conclusions: Can get rid of the first paragraph. Conclusion #3 not supported by the data.
Reviewer 2 Report
Comments and Suggestions for Authors
Abstract
It is preferable to mention the analysis method(s) in the Abstract. Possibly the most important, if there are sever
I would recommend the authors to indicate the practical scales of the obtained results.
Keywords: Please, arrange the keywords alphabetically.
Introduction
The text is too long and verbose (for example lines 127-137), and does not clearly state the research question, the hypotheses, and the significance of the study. It would be better to simplify and shorten the text, and use more specific terms to describe the variables and the outcomes. It would also be helpful to use headings and subheadings to organize the text into logical sections, such as context, problem statement, research question, and significance.
Please, explain in detail your hypothesis and predictions.
I think that you should these important reference as examples to support your sentence:
Nazari, N., Bahmani, M., Kahyani, S., Humar, M., & Koch, G. (2020). Geographic variations of the wood density and fiber dimensions of the Persian oak wood. Forests, 11(9), 1003.
Material and methods
Please, add more details on the statistical analysis.
Results and discussion
It necessary to compare the results with the similar paper research.
Comments on the Quality of English Language
Moderate editing of English language required
